# Professionals' experiences of what affects health outcomes in the sick leave and rehabilitation process—A qualitative study from primary care level

Märit Löfgren[1,2]*, Karin Törnbom[1,3], Daniel Gyllenhammar[4], Lena Nordeman[2,5], Gun Rembeck[1,2,6], Cecilia Björkelund[1,7], Irene Svenningsson[1,8], Dominique Hange[1,7,9]

1 Primary Health Care, School of Public Health and Community Medicine, Institute of Medicine, The Sahlgrenska Academy, University of Gothenburg, Gothenburg, Sweden, 2 Region Västra Götaland, Research, Education, Development & Innovation Primary Health Care, Research, Education, Development & Innovation Center Södra Älvsborg, Sweden, 3 Department of Social Work, University of Gothenburg, Gothenburg, Sweden, 4 Department of Technology Management and Economics, Centre of Healthcare Improvements, Chalmers University of Technology, Gothenburg, Sweden, 5 University of Gothenburg, Sahlgrenska Academy, Institute of Neuroscience and Physiology, Sahlgrenska Academy, University of Gothenburg, Gothenburg, Sweden, 6 Regional Health, Youth Guidance Center, Borås, Sweden, 7 Region Västra Götaland, Research, Education, Development & Innovation Primary Health Care, Sweden, 8 Region Västra Götaland, Research, Education, Development & Innovation Primary Health Care, Research, Education, Development & Innovation Center Fyrbodal, Sweden, 9 Region Västra Götaland, Research, Education, Development & Innovation Primary Health Care, Research, Education, Development & Innovation Center Skaraborg, Sweden

* marit.lofgren@ptj.se

## Abstract

### Objective

To explore frontline employees' experiences of how to create a purposeful sick leave and rehabilitation process (SRP) with the best interest of patients' long-term health in focus.

### Methods

Qualitative design based on focus group interviews in a primary care context in Region Västra Götaland, Sweden. Strategically selected professionals from different SRP organizations discussed sick leave outcomes and the rehabilitation process. Analysis was performed with Systematic text condensation.

### Subjects

General practitioners (n = 6), rehabilitation coordinators and/or healthcare professionals from primary healthcare (n = 13), caseworkers from the Social Insurance Agency, the Employment Agency, and Social Services (n = 12).

### Results

The outcome of the SRP was described to depend upon the extent to which the process meets patients' bio-psycho-social needs. Aspects considered crucial were: 1) early bio-

**Data Availability Statement:** Complete interview data cannot be made publicly available for ethical

and legal reasons according to the Swedish regulations of the "Act concerning the Ethical Review of Research Involving Humans (2006:460)" (https://www.kliniskastudier.se/english/for-researchers/laws-regulations/act-concerning-ethical-review-research-involving-humans–.html) and the Swedish Ethical Review Authority (https://etikprovningsmyndigheten.se/en/). Public availability would compromise participant confidentiality or privacy. Upon request, interview transcripts can be made available after removal of details that may risk the confidentiality of the participants. To access such data, please contact the University of Gothenburg, Sahlgrenska Academy, Institute of Medicine, Department of Public Health and Community Medicine/Primary Health Care, Box 454, (generalpractice@allmed.gu.se), 40530 Gothenburg, Sweden or the author Dominique Hange (dominique.hange@allmed.gu.se).

**Funding:** The study was financed with grants from the Region Västra Götaland, Sweden, and The Kamprad Family Foundation for Entrepreneurship, Research & Charity. The study sponsors had no role in the collection, analysis, and interpretation of data, or the writing of this report, or the decision to submit the article for publication. All authors were independent from the funders.

**Competing interests:** The authors have declared that no competing interests exist.

psycho-social assessments, including medical specialist consultations when needed, 2) long-term realistic planning of sick leave and rehabilitation alongside medical treatment, 3) access to a wide range of early rehabilitative and supportive interventions, including situation-based, non-medical practical problem solving, and 4) trusting relationships over time for all involved professions and roles to maximize process quality and person-centeredness. A gap between the desired scope of the SRP and existing guidelines was identified.

## Conclusion

Interviewees perceived that successful outcomes from the sick leave and rehabilitation process in a primary care context depend on consensus, person-centeredness, and relationship continuity for all involved professions. An extended process scope and relationship continuity for all involved professionals were suggested to improve process outcomes.

## Introduction

According to the Swedish social insurance system, the sick leave and rehabilitation process (SRP) starts when an individual's ability to work is at risk of being impaired, due to illness or injury, and continues until the individual has regained workability (or it is established that the ability to work is permanently reduced) [1]. The SRP aims to regain workability and to achieve a sustainable return to work (RTW).

In Sweden, the socio-economic loss of income due to long-term sick leave amounted to 6 billion Euro in 2022 [2]. Adequate support at an early stage was suggested to have the potential to reduce the need for long-term sick leave, enabling both monetary societal savings and decreased patient suffering [2].

A recent report from the Swedish National Board of Health and Welfare showed that there is insufficient knowledge to define best practices in SRP, both at strategic and operative levels [3]. Compiled evidence from SRP intervention studies showed that early interventions involving collaboration with the workplace are successful strategies for improving workability, but knowledge about which specific interventions have positive effects on workability is limited [4, 5].

The same diagnoses have different consequences for different individuals, particularly in terms of the ability to perform daily activities, fulfill social roles, and participate in work life. Several theoretical models contribute to explaining this phenomenon. For example, health literacy describes the ability to search for, comprehend, and communicate health-related information to make well-balanced and healthy decisions in life [6]. In addition, social insurance literacy describes the extent to which individuals can obtain, understand, and act upon information in a social insurance system [7].

Person-centered care entails shared decision-making based on 1) the patient's narrative about aspects of health and quality of life in the individual (patient's) context, and 2) the professional assessment and evidence-based care [8]. Previous research showed that involving patients in healthcare enables well-informed patients with healthier habits and improved quality of life [9]. Previous studies also showed that person-centered care enabled improved capability for self-management of conditions when compared to usual care [10] and reduced healthcare utilization and care costs [11–13].

A sense of coherence embraces the sense of control and meaningfulness in life by describing to what extent a person finds life comprehensible, manageable, and meaningful [14]. Person-

centered care, health- and social insurance literacy, as well as the sense of coherence, affect the individual's ability to readjust and solve problems. Similar theoretical perspectives are the foundations of the bio-psycho-social model, which adopts a holistic view of the individual. According to the bio-psycho-social model, perception of disease depends not only upon both somatic and psychiatric diagnoses, but also upon social contexts and psychological flexibility [15]. Further, previous research underlined the importance of relationship continuity to enable person-centered care [16, 17], efficient care [10, 18–21], and successful SRP outcomes [22].

Previous research exploring the professional perspective on SRP described how persistent, unclear, and complex combinations of medical conditions (such as long-term physical illness impairing patients' mental health and vice versa) led to prolonged sick leave [23]. Further, non-medical factors affected medical conditions and hindered RTW [24]. The value of supporting the patient to remain active, by returning to meaningful activities and early RTW, was recognized to improve the patient's overall quality of life and mental health [23, 25].

Patients in SRP attempt RTW but discontinue due to their health conditions, feeling chaotic on the inside, fear of overexerting themselves, the individual context, or lack of support from their employers [25–28]. Patients are reluctant to seek help for work difficulties when asking for help with symptom management [29]. However, they request a dialogue about symptom management, treatment options, prognosis, and a plan for RTW [30]. Support aiming for improved self-confidence or help with SRP coordination may also help RTW [25, 26, 31].

This study is part of a larger project with the overall aim of presenting decision support for managing quality-driven change management within SRP. To our knowledge this is the first Swedish study including all professional parties involved in the SRP and thus contributes a holistic and contextual perspective on this area of research.

The purpose of this study was to explore frontline employees' experiences of how to create a purposeful sick leave and rehabilitation process with the best interest of patients' long-term health in focus.

## Materials and methods

### Study design

Consideration of the professional perspective and contextual knowledge is crucial for the validity and importance of results from change management [32]. This study is part of a larger holistic project with the overall aim of presenting decision support for policymakers by answering four predefined research questions:"What are SRP frontline employees' experiences of what affects SRP outcome?","What are SRP frontline employees' experiences of organizational support to apply best practices in SRP?","What potential process improvements do the SRP frontline employees identify based on their experiences?", and"What are SRP frontline employees' experiences of continuous improvement work in the SRP?". This particular study, with its specific background, analysis, discussion, and conclusions, focuses only on the first research question:"What are SRP frontline employees' experiences of what affects SRP outcome?".

A qualitative design was chosen to capture the interviewees' lived experiences, and interviewees were selected strategically to shed light on the research question from as many angles as possible [33]. Data collection was carried out jointly to answer the four predefined research questions, using digital and physical focus group discussions (either/or, no blended groups). According to the focus group methodology, group interaction encourages interviewees to clarify and further explore their statements and perceptions [34, 35]. Focus groups are useful to explore interviewees' experiences of subjects they have in common. The study was reported following the COREQ 32-item checklist for qualitative studies [36].

## Setting and interviewees

The study was conducted in the SRP involving Swedish welfare service and primary healthcare. Interviewees were recruited from different organizations involved in the SRP, i.e., primary healthcare, the Social Insurance Agency, the Employment Agency, and Social Service. The purposive sampling aimed to find interviewees with deep knowledge of the SRP from frequent direct contact with SRP users. In this article, we will henceforth refer to the users as "patients" for clarity, although public service users are referred to differently in different SRP organizations. The sampling also aimed to encompass a broad variety of experiences and perspectives to increase the trustworthiness of the findings by considering organizational affiliation, profession, role in SRP, age, gender, geographical position, and attitude to SRP. To allow geographically dispersed purposive sampling, while at the same time seeking to avoid bias due to the authors impinging on the selection process, rehabilitation coordinators who attended a process manager training course were asked to recommend healthcare interviewees based on a variation in the aforementioned personal characteristics. These rehabilitation coordinators also provided contact persons at the Social Insurance Agency, the Employment Agency, and the Social Services. A total of 41 frontline employees were recommended for participation. Their managers were contacted by e-mail or telephone for permission. The managers of 4 employees did not respond, one declined, and the managers of 36 granted permission. The frontline employees were contacted by telephone to enable them to ask questions about the study. Of 36 persons invited, all wanted to participate, but 4 could not find a suitable time, and one interviewee later declined due to illness. A final sample of 31 was included, consisting of 12 caseworkers (4 each from the Social Insurance Agency, the Employment Agency, and the Social Services, respectively), 6 general practitioners (GPs), and 13 rehabilitation coordinators and/or therapists from primary healthcare. Details of group characteristics are described in Table 1.

## Data collection

Four digital and two physical focus group discussions (the format according to the interviewees' preferences) and each lasting two hours including an introduction and a short break were conducted from September to October 2021. The physical focus group discussions took place on the university premises. The audio recordings were transcribed and pseudonymized, and a code key was established. The code key was stored separately from the transcribed text in a fireproof lockable cabinet at R&D Center Södra Älvsborg.

The focus groups were facilitated by one moderator and one observer, and in the three first focus groups an additional observer followed the conversation, taking notes without participating. All interviews started with recapitulating the aim of the study and letting everyone introduce themselves. Interviewees were encouraged to share all their experiences about the SRP, both positive and negative. To ensure an open-hearted dialogue, group rules of conduct, including a respectful dialogue and moral obligation of confidentiality, were established.

All focus groups included the following main research questions: what affects the SRP outcome, in what ways do we work with SRP, experiences of the service as a whole, organizational support, and potential improvements. The introductory question was "Could you tell us about your experiences of the sick leave and rehabilitation process?". The interviews were semi-structured, although the moderators made sure that discussions stayed focused on answering the research questions (see S1 File for the interview guide). To achieve a deep understanding of interviewees' experiences, the moderators used open-ended questions and probing questions for clarifications and explanations, e.g., "Can you tell me more about that?". Furthermore, the moderators continuously asked clarifying questions not only to ensure having understood the interviewees correctly, but also to explore whether there was an agreement or different

**Table 1. Description of participants (n = 31) in a qualitative study exploring frontline employees' experiences of what affects health outcomes in the sick leave and rehabilitation process (SRP) in a primary care context.**

| Focus group # | Format | Participants' organization, profession, or role | Sex (men/women) | SRP experience </> than 5 years | Number of PCCs or municipalities represented | Size of PCC patient population | Geographic representation | Perspectives represented (all focus groups) |
|---|---|---|---|---|---|---|---|---|
| 1 | Digital | Caseworkers (n = 6): *Employment Agency (n = 2), Social Insurance Agency (n = 2), Social Services (n = 2)* | 0/6 | 1/5 | 3 municipalities | - | Small city (n = 4), urban area (n = 1), sparsely populated area (n = 1) | 1) Work oriented rehabilitation and coordination of SRP cases from the perspective of the Social Insurance Agency, the Employment Service, and the Social Services. 2) General practice, insurance medical responsiblity. 3) Local management, and local multi-professional teamwork, within primary healthcare. 4) SRP coordinator role within primary healthcare. 5) Occupational therapy, physiotherapy, psychotherapy/psychology, psychosocial team triage, care management depression. 6) Work oriented rehabilitation from the perspective of primary healthcare. |
| 2 | Digital | General practitioners (n = 4) | 1/3 | 0/4 | 4 PCCs | 4 100–13 000 | Big city (n = 1), small city (n = 2), sparsely populated area (n = 1) | |
| 3 | Physical | Rehabilitation—coordinator role and/or profession—from primary healthcare (n = 5) | 2/3 | 0/5 | 8 PCCs | 7 400–18 400 | Small city (n = 1), urban area (n = 3), sparsely populated area (n = 1) | |
| 4 | Digital | Rehabilitation—coordinator role and/or profession—from primary healthcare (n = 5) General practitioner (n = 1) | 0/6 | 2/4 | 8 PCCs | 7 600–10 600 | Big city (n = 2), small city (n = 1), urban area (n = 1), sparsely populated area (n = 2) | |
| 5 | Digital | Caseworkers (n = 6): *Employment Agency (n = 2), Social Insurance Agency (n = 2), Social Services (n = 2)* | 1/5 | 2/4 | 3 municipalities | - | Big city (n = 2), small city (n = 2), urban area (n = 1), sparsely populated area (n = 1) | |
| 6 | Physical | Rehabilitation—coordinator role and/or profession—from primary healthcare (n = 3) General practitioner (n = 1) | 1/3 | 2/2 | 4 PCCs | 9 200–11 700 | Big city (n = 1), small city (n = 2), sparsely populated area (n = 1) | |

This table describes the characteristics of 31 frontline employees participating in digital or physical focus group interviews about the SRP. The interviewees came from primary healthcare, the Social Insurance Agency, the Employment Agency, and the Social Services. They represented different professions and perspectives, were mostly experienced (>5 years working within the SRP), but also in the beginning of their working life, and came from different primary care centers (PCCs) and municipalities. The PCCs represented included small and large units and PCCs from both urban and sparsely populated areas.

opinions within the group. Before ending the recording, the interviewees were given the opportunity to present their personal summarized views on what had emerged.

The observer made field notes during the focus groups. The interview strategy, selected questions, and role of the interviewers were discussed and adapted between the interviews to optimize data collection. As the interviewees only needed the introductory question to initiate vivid discussions, the interviewers used the interview guide to make sure all four research questions were addressed. Still, the questions were addressed in different orders in the various focus groups depending on how the discussions went in each group.

The last group conducted only showed variations on previous themes, hence indicating information depth and saturation. After the focus groups the authors contacted two interviewees to clarify certain aspects that first had been difficult to comprehend.

### Ethical consideration

The study was approved by the Swedish Ethics Review Authority (Dnr 2021–01481). All interviewees signed an informed written consent document. The interviewees did not receive any compensation for their participation. The research team had no personal interest in the results.

### Data analysis

Data was analyzed using systematic text condensation according to Malterud [37]. This qualitative analysis method was chosen because of its pragmatic, holistic, explorative, and reflexive approach, which we believe suited our overall aim of presenting decision support for managing quality-driven change management within SRP, considering the perspectives of different professional parties involved in SRP, and our holistic approach.

Data transcriptions were used for the analysis. Interviewees were pseudonymized to ensure their anonymity. In the first step, both authors DG and ML read the interviews in their entirety, independently identifying preliminary themes. The preliminary themes, which were similar despite mirroring the researchers' different paradigms, were then discussed to reach a consensus about themes for the continued analysis. In the second step, meaning units were identified independently by DG and ML, then discussed and grouped by the preliminary themes. The coding process was done using the NVivo program [38]. The content of the preliminary themes and the researchers' understanding of delimitations changed during the process. Also, the researchers' different research paradigms, i.e., the "Social Welfare System" perspective of the management scholar and the "What's best for the patients?" perspective of primary healthcare, emerged more clearly as the analysis proceeded. To accommodate both scientific fields, two sets of codes, each with thematic sub-codes (thus integrating the third step of the analysis), were agreed upon. In the continued analysis, DG and ML managed separate analysis processes, yet participated in each other's processes to provide interdisciplinary advice and to make sure both analyses stayed true to the data. The management-oriented report was recently published [39].

As a continuation of the third step of the analysis from a primary healthcare perspective, meaning units were sorted into thematic sub-groups within three codes: what affects SRP outcome, the gap between desired process outcome and reality, and a systems perspective on realizing a person-centered and efficient SRP.

In the fourth step, groups of condensates were refined as a coherent analytic text and recontextualized using quotes.

## Results

The main finding was that interviewees perceived that the primary factor influencing the outcome of the SRP is the extent to which the process meets patients' bio-psycho-social needs by enabling consensus, person-centered and realistic SRP plans, participation in meaningful activities, and relationship continuity. Patients' bio-psycho-social needs encompass the needs that patients are aware of, and therefore spontaneously express, and their additional needs that the interviewees, with their professional expertise, may identify, and address in dialogue with the patients. Table 2 summarizes the analysis structure and the key findings.

### Creating consensus

**Causes of ill-health.** Interviewees stated that many factors beyond patients' medical conditions influenced the outcome of the SRP. Reaching a consensus among professionals and

**Table 2. Analysis structure and summary of frontline employees' experiences of what affects health outcomes in the sick leave and rehabilitation process (SRP) in a primary care context.**

| Aim | Code groups | Subgroups | Condensed findings | Implications on the scope of SRP |
|---|---|---|---|---|
| *To explore frontline employees' experiences of how to create a purposeful sick leave and rehabilitation process with the best interest for patients' long-term health in focus* | **Creating consensus** | Causes of ill-health Opportunities available Assessment | The room for action in SRP was perceived to be determined by diagnoses and treatment options, but also by rules and regulations and the individual context. The quality of the bio-psycho-social assessment, including specialist consultations when needed, was critical for consensus around a person-centered problem description. | *Interviewees perceived that successful outcomes from the sick leave and rehabilitation process in a primary care context depend on consensus, person-centeredness, and relationship continuity for all involved professions. An extended process scope and relationship continuity for all involved professions was suggested to improve process outcomes.* |
| | **A realistic rehabilitation plan** | Long-term perspective Adequate timing Overcoming obstacles in daily life Multi-professional support | A long-term plan for sick leave and rehabilitation needed to provide clear and satisfying, individualized information for the patient about how to achieve sustainably optimized health and function over time. | |
| | **Participation in meaningful activities** | Individually meaningful Preserving skills Accessibility Person-centered dialogue | Remaining as active as possible, striving to preserve and develop skills despite illness, was described to benefit both well-being and long-term abilities. A wide range of accessible rehabilitative and supportive interventions, including situation-based practical problem-solving, was needed to accommodate realistic individual rehabilitation planning. | |
| | **Trusting relationships over time** | Person-centeredness Quality of care | Interviewees argued that continuous therapeutical relations over time contributed to better understanding of the patients as individuals, their medical history, and their work capacity. Further, it was perceived to have positive effects on quality of care, legal decisions, and person-centeredness. | |

This table describes the analysis structure and summarized results from the first of three qualitative analyses based on data collected with focus group interviews involving 31 frontline employees in the sick leave and rehabilitation process. Interviewees were from primary healthcare, the Social Insurance Agency, the Employment Agency, and the Social Services. The analyses were intended to provide decision support for managing quality driven change management within SRP.

with the patient regarding the cause of the patient's ill health was described as an important prerequisite for achieving a coordinated, person-centered SRP and crucial for process outcome.

A combination of medical and non-medical factors, which together affect the well-being and work capacity of patients, was considered to be standard rather than an exception in SRP. Interviewees perceived this was the case regardless of whether patients entered the SRP through primary care (seeking treatment for illness) or through the Employment Service/Social Services (unemployed individuals where ill health affects employability).

> "...it is very easy to fall into a narrow diagnostic way of thinking, where all reduced ability is connected to a diagnosis. However, that is not always the case; it's actually very rare that reduced work capacity is solely related to a diagnosis." *GP 13–3*

**Opportunities available.** Consensus on expectations and beliefs related to opportunities for medical treatment, SRP regulations, rights and obligations, and a shared view on access to SRP support and work adaptations was considered highly important. Patients' expectations of help from healthcare and society and their beliefs and fears varied, thus influencing their perspectives on the need for sick leave and rehabilitation.

Understanding each patient's situation was described as a central element in making patients receptive to and motivated for healthcare and rehabilitation. Interviewees underscored that information and explanatory models need to be individually adapted to the patient's unique context, as "SRP patients" are a very heterogeneous group with different bio-psycho-social conditions and needs. These differences were described as involving variations in symptoms and diagnoses, in cognitive ability, and in adaptability. Also, patients were situated in different workplaces with different demands and in different contexts with various socioeconomic conditions and support.

**Assessment.** The often complex, individual needs of patients were described as placing great demands on the assessment process, which sought to enable problem understanding and consensus. Interviewees agreed that when primary care (together with the patient and/or the Social Insurance Agency/Employment Service/Social Services) deems that the answer to a medical question is crucial for the patient's ongoing rehabilitation, the question needs to be satisfactorily addressed. In cases where primary care lacked the expertise to provide satisfactory answers to medical questions, access to consultative assessments within specialized care was considered central to achieving inter-professional and patient consensus. The interviewees strongly emphasized the importance of specialist care providing clarity to make progress in challenging cases;

". . .Sometimes a more thorough examination of someone's (the patient's) health status may be necessary [. . .] Either because one requires an intervention from psychiatry. . . a treatment, or to find out that, no, this is not the issue (something initially assumed), to not end up in a sort of status quo, where one doesn't know how things stand." *Caseworker from the Social Insurance Agency 27–5*

## A realistic rehabilitation plan

**Long-term perspective.** The interviewees consistently stressed that patients need a predictable, long-term plan not only for the SRP, but also during the RTW phase. This plan should provide clear and satisfying information about how the practical and tangible process of RTW should be carried out. Interviewees highlighted the importance of bringing about solutions that could maintain health in a long-term perspective, to enable patients to dare to RTW.

"I find that. . . if one doesn't have a clear image of whether it's a year, or is it a month I'm going to be on sick leave? . . ..and what happens afterwards? (an uncertainty). Then one tends to hold on to one's sick leave." *Rehabilitation 22–5 (coordinator role)*

**Adequate timing.** Patients' pre-understanding—their expectation of help from healthcare and society, their beliefs, and their fears—needed to be considered early in the SRP to bridge the gaps between what patients had initially envisaged, the realistic room for action, and what is considered best for the patients in the long run. The interviewees asserted that specific information, coupled with a plan for individually tailored support measures, could be pivotal in reinforcing the patient's possibilities for RTW. Adequate timing of activities was considered

crucial in helping patients adhere to their SRP plan, remain hopeful, and limit negative thoughts.

"...They [referring to patients in SRP] often need information, it's often concrete information they need. What has happened at the workplace? What does it look like if you're going back? And what does the plan look like then? That the employer makes this planning, as they should do per law; a return-to-work plan and all that. Because if all this is done, it dramatically eases the situation and the anxiety (of the patient) decreases." *Rehabilitation 16–4 (coordinator role)*

**Overcoming obstacles in daily life.**   Based on the collective experiences of the interviewees, the consensus was that the patients' capacities to manage their ill health and life in general significantly influenced the progression and outcome of their SRP. The interviewees stated that patients' individual abilities and circumstances always needed to be taken into consideration.

The interviewees perceived that supporting patients to extricate themselves from an unhealthy situation enhanced successful SRP outcomes. It was perceived that poor health reduced patients' resources making it difficult for them to break free from a downward spiral, despite being well aware of their responsibility in the situation. As an example, it was explained that patients who were balancing on the edge between a normal reaction to severe stress and conditions such as depression or anxiety could encounter difficulties in seeking and securing new employment.

Work-oriented rehabilitation was perceived to contribute to a deeper understanding of the patient's health problems, functional and activity capacity, as well as the need for support and adjustments. The interviewees described that increased and shared understanding made it easier for all stakeholders to reach a consensus on continued efforts.

**Multi-professional support.**   It was perceived that an SRP plan needs to strike a balance between regard for patients' personal preferences, the needs identified by the professionals, and the boundaries established by existing regulations. Access to coordinated multi-professional support to overcome obstacles along the way was described as a crucial prerequisite to enhance patients' sense of security and success during their rehabilitation.

"I see it this way, as we (speaking of "we" as various professions) come from different paradigms, and there... it is very fruitful to blend paradigms... meaning both a medically grounded paradigm and a humanistic paradigm under the same umbrella... one can make much more refined assessments from multiple perspectives." *Rehabilitation 14–1 (psychologist profession)*

The interviewees highlighted that patients in SRP often searched and asked for the special expertise that they, in their specific role and profession, could provide them with. For example, GPs lifted their role in planning medical investigations to facilitate diagnosis and to plan treatments together with the patient, while rehabilitation coordinators emphasized the importance of patients reaching out to them, to understand the regulatory framework around the SRP and to strategically plan their RTW. Further, interviewees from rehabilitation and psychosocial teams emphasized the importance of their engagement in giving patients access to medical rehabilitation, patient education, and self-care support, as a way of avoiding the medicalization of normal reactions, such as personal crises or a perceived overload from work. Finally, caseworkers from the Social Insurance Agency, the Employment Agency, and the Social Services

strongly emphasized the benefit of being able to offer work-oriented rehabilitation early in the process.

## Participation in meaningful activities

**Individually meaningful.** The interviewees perceived it as crucial for patients' health outcomes that the goals set in the SRP were experienced as meaningful by the patients. Examples of meaningful goals mentioned included increased physical function, improved ability to participate in social contexts and at work, as well as goals that derive their meaning from an individual context.

**Preserving skills.** Interviewees unanimously stated that the long-term best outcome for patients with long-term illness or disabilities was for them to be able to remain active based on their current abilities, rather than being on full-time sick leave. They described how patients benefit from getting out of the home and participating in social contexts–both to feel better and to avoid losing their social skills during the sick leave period.

> "It is immensely important for a person [as a patient on sick leave] to engage in some form of activity. To get out of the house for an hour or a few hours per week. Not just to meet doctors or healthcare contacts, but to be welcomed and be part of a different social context." *Caseworker from the Employment Agency 6–2*

**Accessibility.** The interviewees argued that to enable a person-centered rehabilitation plan and achieve a balance between rest and activity, there needed to be a wide range of easily accessible rehabilitative interventions. Examples of activities mentioned included workplace adjustments, counselling support during life crises, assistance in solving practical problems, opportunities for work-oriented training, and (undemanding) social rehabilitation.

**Person-centered dialogue.** It was considered important to involve the patients in their healthcare through dialogues about expectations, concerns, and perceived needs to cope with the challenges of living with a chronic or long-term illness/vulnerability or recurring health issues. An active dialogue with patients was fundamental, not only to ascertain whether patients felt well-informed, but also to address any practical issues that could hinder their RTW or pose a risk of relapse into illness. According to the interviewees, there are numerous solutions to bring about improvements that may create a ripple effect.

> ". . .one [in the process of sick leave and rehabilitation] needs someone else to say, 'You know you can make changes, right?' It is possible to make changes, even though you [as a patient] may feel. . . deeply depressed. Sometimes, it may be appropriate to make a change in the fundamental situation to help the patient move forward." *Rehabilitation 14–1 (psychologist)*

## Trusting relationships over time

**Person-centeredness.** A clear and continuous professional responsibility over time was considered to be an important factor for achieving positive health outcomes in the SRP. Continuity was described as a prerequisite for meeting patients' need for security, through trusting relationships and easy communication channels. Furthermore, continuity of therapeutic relations was considered necessary to get to know the patients and understand what is important and meaningful to them as individuals to provide person-centered care. As many health problems result in recurrent impairment of work capacity, it was argued that person-centered continuity of care is desirable, both within the same sickness period and in cases of relapse.

"If it's someone you trust who conveys the assessment that 'sick leave might not be a good idea here'. . . It's easier for you [as a patient] to accept that perspective, I think, compared to if. . . anyone else says the same thing. But above all, it ensures that the interventions are appropriate if you [as a GP] are not just coming in from the sidelines to make a sick leave assessment. I believe that it saves resources in general (referring to continuous contacts)." *GP 13–3*

**Quality of care.** Continuity of therapeutic relations was considered critical to reach a consensus among professionals and the patient both regarding the bio-psycho-social problem description, a realistic rehabilitation plan, and participation in meaningful activities.

"I'm thinking about the communication in the team. That we speak the same language, that we mean the same thing and that we agree on things. . . that we show together in the team that we. . . we want to help the patient move forward but also that we say the same things or mean the same things when we talk to the patient, so it becomes clear. Again, for the patient's sense of security, whether we are saying uncomfortable things or not. . ." *Rehabilitation 22–1 (coordinator role)*

The importance of continuity of care was emphasized by both physicians, rehabilitation coordinators, and administrators. It was linked to both legal security and increased quality of care, as well as improved inter-professional collaboration over time, with a focus on the best interests of the patients.

"The need for sick leave cannot be proven, only made probable. It requires mutual honesty and openness between the physician and the patient, which presupposes trust. A patient who does not trust the physician, or a physician who does not trust the patient leads to a less reliable assessment. Continuity breeds trust and facilitates this openness, I believe." *GP 13–3*

## Discussion

Our results indicate that to optimize outcomes from SRP, the process needs to embrace and facilitate the management of both medical and non-medical factors affecting health and work capacity by incorporating the following aspects: 1) creating consensus about causes of ill-health and opportunities available through early bio-psycho-social assessments, including medical specialist consultations when needed, 2) a realistic rehabilitation plan for regaining and sustaining health through timely multi-professional rehabilitation, practical support to overcome obstacles in daily life, and RTW, alongside medical treatment, 3) access to a wide range of rehabilitative and supportive interventions to enable participation in individually meaningful activities and preserving skills, and 4) trusting relationships over time to maximize process quality and person-centeredness.

An important finding from the theme *Creating consensus*, was that the interviewees, regardless of their profession and role, strongly agreed on the perception that reduced work capacity rarely can be explained solely by medical diagnoses. The impact of living with a functional impairment (which may be associated with multiple diagnoses) was described as depending on the social context as well as individual coping strategies, and together these affect the outcomes in work rehabilitation. Thus, non-medical factors were explained to have a significant influence on an individual's working capacity. These findings are consistent with previous research on differences in health literacy [6, 7], sense of coherence [14], and the bio-psycho-

social model [15]. It is also in line with previous qualitative research on the professional perspective on SRP [23–25], which together highlight the importance of adopting a bio-psychosocial perspective. Therefore, we argue that the findings of this study offer further grounds to challenge the prevailing practice in SRP exclusively emphasizing diagnosis and illness.

An early bio-psycho-social assessment of what might be causing the poor well-being and impaired workability, and conceivable options, was seen as central to enabling consensus about adequate actions. Wisely chosen early medical assessment, including specialized healthcare when needed, was argued as a prerequisite for primary healthcare in order to provide credible answers to both patients' and SRP professionals' questions, adequate advice regarding available treatments, and a basis for a realistic plan for the SRP. Consultative support from specialist healthcare was valued even in cases where further specialist interventions are not beneficial. This result confirms previous knowledge on the importance of perceiving life difficulties as comprehensible [14]. It is also consistent with the concepts of health literacy [6] and social insurance literacy [7], which highlight the central role of knowledge in making informed decisions. Our findings indicate that if the patients do not fully understand their symptoms, or if they believe they are being withheld from specialist medical treatment, there is a risk of patients experiencing a loss of control and acting counterproductively by not following advice on functional rehabilitation or adjustments in life. Based on the results, we argued that the same reasoning is valid for SRP professionals. They need to understand the patient's problem to maintain control and act constructively to help SRP patients.

The result suggested the importance of professionals initiating patient dialogues of a more in-depth nature to understand the patient's perspective in SRP. For example, exploring the patient's ideas, concerns and expectations about symptoms and recommended actions concerning RTW was vital, rather than solely engaging in conversations about the right or not to sick-leave benefits. This finding is in line with both existing guidelines for SRP [40, 41], person-centered care [8] and the patient-centered consultation model [42, 43]. The need for a more substantive dialogue is also corroborated by research describing the complex and sensitive nature of SRP from a patient perspective [25–31].

Based on the theme, *Creating consensus*, we have drawn three conclusions about strategies that should coexist in the SRP: 1) it is important to listen and take the patient's symptoms seriously, regardless of medical or non-medical causes—the patient's suffering can be equally significant, real, and addressable in all cases; 2) sufficiently answering patients' health questions is key to their trust, compliance to both medical and non-medical advice, and reorientation in life; and 3) in the absence of an evidence-based medical intervention, it is important to act based on the best available evidence. If there are no specific medical treatment options available, functional and social rehabilitation is considered to be the best approach for the patient.

A main finding from the theme *A realistic rehabilitation plan* was the importance of early concretization of a plan for sustainable RTW with due consideration of the patient's health needs. Coordinated long-term planning and problem-solving were perceived as vital for achieving consensus on problem analysis, management, and responsibilities between patients and professionals. Further, patient involvement in designing the SRP plan was described as having a positive influence on its outcome, avoiding unrealistic demands and promoting consensus. This perception of a connection between health outcomes and patient involvement in care planning aligns with previous research and guidelines emphasizing its significance [8, 42, 43].

Underscoring the importance of a realistic SRP plan, considering the patient's well-being, resources, and life situation, and ensuring that goals and activities are meaningful to the patient, aligns with a sense of coherence [14], which emphasizes the importance of both a sense of manageability (realistic planning, necessary support as needed) and meaningfulness (based on what is important to the patient). The significance of person-centered approaches in

SRP is also consistent with existing guidelines [8, 40, 41] and with research on the patient perspective on SRP [25–31].

The interviewees referred to SRP as a longitudinal process that in practicality starts before the patient goes on sick leave (with, for example, support for achieving work-life balance to prevent the need for sick leave) and needs to continue also after RTW. Thus, patients with chronic conditions require ongoing follow-up and support for a sustainable work capacity. The notion that SRP considerations need to continue after the sick leave period differs from previous recommendations [1].

Among interviewees, there was strong consensus that the long-term best approach for all SRP patients was to remain as active as possible, including in work or work-oriented rehabilitation, alongside medical interventions. This finding, from the theme *Participation in meaningful activities* is in line with previous research [4, 5, 23, 25] and consistent with guidelines for SRP [40, 41] that advocate for integrating medical rehabilitation as early as the patient's condition allows, rather than providing it as a subsequent, completely parallel, or separate process. However, the current study also emphasized the importance of early social rehabilitation through, for example, work-oriented rehabilitation to enable participation in social contexts, in harmony with a sense of coherence [14]. Further, the impact of supporting patients to enable adherence to an SRP plan, for example by means of easily accessible psychosocial support and practical support in managing daily life during challenging periods, was highlighted. Compared to existing SRP guidelines [40, 41], the results of this study more clearly emphasize that patients need individually tailored non-medical practical support, even outside the workplace.

Based on combined findings from the first three themes, we concluded that an SRP plan should comprise a bio-psycho-social assessment, medical care, rehabilitation, social and activity-based interventions, and efforts to solve practical non-medical problems. The plan should also include a clear realistic plan for RTW and the length of sick leave, with the aim to enable sustainable workability.

The fourth theme of the present study, *Trusting relationships over time*, suggested that maintaining relationship continuity among all professions involved in a patient's SRP, both within primary healthcare, across different agencies and across different sick leave cases over time, contributes to the quality of care and reduces the need for sick leave. These results align with previous research that demonstrated positive effects of relationship continuity on patient satisfaction, person-centeredness, healthcare outcomes, and the reduction of healthcare utilization and sick leave [16–22].

However, the Swedish standard for person-centered care does not explicitly highlight the notion that relationship continuity is a prerequisite for person-centered care [8], nor do guidelines for SRP [40, 41]. We concluded that the complexity of a person-centered SRP in primary care seems to place greater demands on relationship continuity than person-centered episodic inpatient care or care focusing on a single diagnosis. Our findings suggest that there is a need to expand the concept of person-centeredness in the context of SRP and primary healthcare, to embrace relationship continuity.

In Sweden, only 35% of the adult population have a designated GP or nurse at their healthcare center, whereas in comparable countries, the corresponding figure ranges between 80–98% [44]. There is a strategic political will to improve relationship continuity in Swedish primary healthcare [45], but the political will has not yet permeated the operational level [46]. Furthermore, discussions about relationship continuity have been limited to having a designated GP or care contact person, rather than considering relationship continuity as a general principle for all professions involved in a patient's care, as suggested by our findings from the SRP.

## Considerations for a purposeful sick leave and rehabilitation process

The results of this qualitative study indicate that there is a need to redefine the scope of SRP. Early consideration of non-medical factors affecting health and workability, alongside medical care, was suggested to enable better SRP outcomes than exclusively emphasizing diagnosis and illness. Notably, a broadened SRP scope would exceed primary healthcare responsibilities and require new interventions and ways to collaborate to be jointly developed by primary healthcare and other SRP actors.

Maintaining relationship continuity among all professions, across different agencies, and across patients' different sick leave cases over time was highlighted as a strategy for facilitating problem understanding, person-centered SRP planning, and as a prerequisite for consensus and stability, thus contributing to the quality of care and reducing the need for sick leave.

Comparing our findings to the guidelines for SRP and person-centered care [8, 42, 43], we found the existing guidelines do not fully comply with the results of this study. To bridge the gap, we suggest investigating the possibility of an extended SRP scope, focusing on enabling an agile, person-centered and active process and embracing relationship continuity in the definition of person-centeredness in the context of SRP and primary healthcare.

## Strengths and limitations

The current study represented a holistic approach by including all professional parties, yielding a rich and nuanced, yet largely common view of the SRP process. The research design provided a deep understanding of the SRP process. It captured details, different perspectives, common grounds, and those areas that engaged interviewees the most, both emotionally and opinion-wise. All interviewees spoke from their own experiences, enabling us to interpret the phenomena in its relevant context and to understand the complexity within the SRP process.

Digital and physical focus groups were equally rich in content, lively, and engaged. There was a minor difference in the quality of data collected overall. Nevertheless, we noted that it was much more strenuous for the interviewers to conduct the digital focus groups, as the lack of direct interaction between people required the interviewers' compensatory effort.

Unlike quantitative research, qualitative findings do not prove hypotheses, as participant selection is strategic rather than random, and conclusions are drawn from the statements of a select few individuals. Instead, qualitative research methods aim to uncover the feelings, beliefs, and experiences of a particular target group regarding a specific topic, providing insight into their behaviors. When utilized thoughtfully, the descriptive findings from qualitative research play a critical role in making contextually relevant decisions and laying the groundwork for subsequent quantitative studies.

The interviewees' statements regarded diagnoses and conditions that are either difficult to verify objectively, or whose progress is unpredictable, recurrent or chronic, such as common mental disorders and non-traumatic musculoskeletal pain, which together represent 2/3 of sick leave cases exceeding 14 days in Sweden [47]. We concluded that our findings are relevant to consider in situations when the medical condition is complicated, when the individual context is complex, or when the medical benefit of sick leave is unpredictable.

Interviewees' experience in their fields varied, but the GPs included in the sample were more experienced than average GPs, as they were medical directors of the local SRP work. However, this was compensated for by including rehabilitation coordinators and caseworkers who had experience from working with different GPs. As SRP patients are common in primary healthcare and at other agencies, all interviewees were familiar with the process.

Interviewees were selected based on recommendation from external parties, to avoid author bias or convenience sampling. Consistently following recommendations considering the

purposive sampling was a priority in the few cases where there were prior professional connections between an author and invited interviewees. These connections were unlikely to have biased or influenced the data collection.

The authors represented different research fields. Their different perspectives were helpful both in understanding the interviewees' perceptions from different paradigms and in maintaining the interdisciplinary approach of the study [48].

## Conclusions

According to the interviewees, successful outcomes from the sick leave and rehabilitation process (SRP) depend upon enabling consensus around early bio-psycho-social understanding of the patients' health problems, a long-term plan for person-centered care that includes activities targeting non-medical factors contributing to ill health, and patients participating in social and work life (with respect to their impaired health). Relationship continuity for all involved professions and roles was described as a prerequisite for a person-centered and integrated process.

The study identified a possible gap between the desired scope of the SRP and existing guidelines and strategies. The authors suggested investigating an extended scope and relationship continuity for all involved professionals to enable a person-centered SRP and improved outcomes in the context of primary healthcare.

## Supporting information

**S1 File. Interview guide.**
(DOCX)

## Acknowledgments

The authors are grateful to all participating interviewees and to their managers who let them participate. We are also grateful to Åsa Radl, Eva Hällås, and Malin Rydén Mölholm, who helped with the purposive sampling.

## Author Contributions

**Conceptualization:** Märit Löfgren, Karin Törnbom, Daniel Gyllenhammar, Lena Nordeman, Gun Rembeck, Cecilia Björkelund, Irene Svenningsson, Dominique Hange.

**Formal analysis:** Märit Löfgren, Karin Törnbom, Daniel Gyllenhammar.

**Funding acquisition:** Märit Löfgren, Dominique Hange.

**Investigation:** Märit Löfgren, Karin Törnbom, Daniel Gyllenhammar.

**Methodology:** Märit Löfgren, Karin Törnbom, Daniel Gyllenhammar, Lena Nordeman, Gun Rembeck, Cecilia Björkelund, Irene Svenningsson, Dominique Hange.

**Project administration:** Märit Löfgren, Dominique Hange.

**Resources:** Märit Löfgren, Daniel Gyllenhammar.

**Supervision:** Karin Törnbom, Lena Nordeman, Gun Rembeck, Cecilia Björkelund, Irene Svenningsson, Dominique Hange.

**Visualization:** Märit Löfgren.

**Writing – original draft:** Märit Löfgren, Karin Törnbom.

**Writing – review & editing:** Karin Törnbom, Daniel Gyllenhammar, Lena Nordeman, Gun Rembeck, Cecilia Björkelund, Irene Svenningsson, Dominique Hange.

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
