## [Decision Letter · Decision Letter 0]

25 Mar 2024

PONE-D-23-40927Professionals’ experiences of what affects health outcomes in the sick leave and rehabilitation process - a qualitative study from primary carePLOS ONE

Dear Dr. Löfgren,

Thank you for submitting your manuscript to PLOS ONE. After careful consideration, we feel that it has merit but does not fully meet PLOS ONE’s publication criteria as it currently stands. Therefore, we invite you to submit a revised version of the manuscript that addresses the points raised during the review process. Please submit your revised manuscript by May 09 2024 11:59PM. If you will need more time than this to complete your revisions, please reply to this message or contact the journal office at plosone@plos.org. Please include the following items when submitting your revised manuscript:A rebuttal letter that responds to each point raised by the academic editor and reviewer(s). You should upload this letter as a separate file labeled 'Response to Reviewers'.A marked-up copy of your manuscript that highlights changes made to the original version. You should upload this as a separate file labeled 'Revised Manuscript with Track Changes'.An unmarked version of your revised paper without tracked changes. You should upload this as a separate file labeled 'Manuscript'.

We look forward to receiving your revised manuscript.

Kind regards,

Muhammad Shahzad Aslam, Ph.D.,M.Phil., Pharm-D

Academic Editor

PLOS ONE

Reviewers' comments:

Reviewer's Responses to Questions

**Comments to the Author**

1. Is the manuscript technically sound, and do the data support the conclusions?

Reviewer #1: Yes

Reviewer #2: Yes

2. Has the statistical analysis been performed appropriately and rigorously? 

Reviewer #1: N/A

Reviewer #2: Yes

3. Have the authors made all data underlying the findings in their manuscript fully available?

Reviewer #1: Yes

Reviewer #2: Yes

4. Is the manuscript presented in an intelligible fashion and written in standard English?

Reviewer #1: Yes

Reviewer #2: Yes

5. Review Comments to the Author

Reviewer #1: Abstract

Professional actors could be a confusing term, the word actors could be misinterpreted.

Include an explanation of why these particular professionals were recruited.

Describe what type of analysis that was used.

Method

Describe why this method was suitable for this study, why was it selected instead of other qualitative approaches?

Did you reflect on differences in the interviews between digital and physical focus groups?

How was the interview guide developed?

The research questions are presented for the first time under the section analysis. Why are these research questions not presented earlier? And how were they developed in relation to the research aim?

In the article it is stated:

“ Therefore, the research group decided to report the analysis in a series of three different articles.”

This is unclear to me, will the same analysis be presented in three different articles? If so, I think that is a problem. The data consists of six interviews. In my view this is not a lot of data to justify different articles. And how are the findings presented in this article related to the findings in the other two articles? I suggest that the analysis form the six interviews is presented in this article.

Results

What do you mean by ‘meets patients conscious and unconscious needs? What needs are unconscious, and who have been able to identify these needs? I think the word unconscious is problematic. Could it be explained in a different way?

I would suggest that you use headings in your result section to make clear what code group and you are presenting.

The subgroups are not always clear in the analysis. I would suggest that you revise the analysis so that the subgroups can be easily found in the result section. Make it very easy for the reader to follow and understand your results.

Discussion

The fist paragraph in the discussion presents what need to be done to optimize outcomes form SRP. Are these the same as your code groups, or are they somewhat different? For example Early ‘early bio-psycho-social assessments’ appears to be one sub group in the first code group (or I may have misunderstood this). It would make it more easy to understand if you connect the summary of your results to your code groups.

The first main finding presented is not one of the code groups. This is a bit confusing, or is it Is this relating to the first code group, creating consensus?

“A main finding of this article was that the interviewees, regardless of their profession and role, strongly agreed on the perception that reduced work capacity rarely can be explained solely by medical diagnoses.”

There are very few limitations included. Could it be a limitation that some of the interviews were digital and some live? I would suggest that the authors reflect on this under limitations.

Reviewer #2: The manuscript was well written and informative, the readers will be able to learn a methodology of a qualitative study. If possible to shorther the lenght of manuscript the results may be shorthen.

The interpretation of results may be added to the discussion no need to have a separated section and more focust and expamd the link with SRP in order to highlight the application of the result.

6. PLOS authors have the option to publish the peer review history of their article (what does this mean?). If published, this will include your full peer review and any attached files.

Reviewer #1: **Yes: **Kristina Aurelius

Reviewer #2: No

---

## [Author Response · Author response to Decision Letter 0]

15 May 2024

Dear Reviewers,

Thank you for reviewing the Manuscript ID PONE-D-23-40927 "Professionals’ experiences of what affects health outcomes in the sick leave and rehabilitation process - a qualitative study from primary care". We are grateful for the time and effort that you, and the Editorial Board, have used to examine our manuscript. 

We have carefully read your comments, and addressed all the points raised point-by-point. You will find our rebuttal letter in the separate file labeled "Response to Reviewers". You will also find our manuscript in a marked-up copy that highlights changes made to the original version, and an unmarked version without tracked changes. 

We hope that this revision meets your anticipations. If you want further changes in the manuscript, please notify us. 

For the authors

Yours sincerely,

Märit Löfgren

---

## [Decision Letter · Decision Letter 1]

11 Jun 2024

Professionals’ experiences of what affects health outcomes in the sick leave and rehabilitation process - a qualitative study from primary care level

PONE-D-23-40927R1

Dear Dr. Löfgren,

We’re pleased to inform you that your manuscript has been judged scientifically suitable for publication and will be formally accepted for publication once it meets all outstanding technical requirements.

Kind regards,

Muhammad Shahzad Aslam, Ph.D.,M.Phil., Pharm-D

Academic Editor

PLOS ONE

Additional Editor Comments (optional):

Reviewers' comments:

Reviewer's Responses to Questions

**Comments to the Author**

1. If the authors have adequately addressed your comments raised in a previous round of review and you feel that this manuscript is now acceptable for publication, you may indicate that here to bypass the “Comments to the Author” section, enter your conflict of interest statement in the “Confidential to Editor” section, and submit your "Accept" recommendation.

Reviewer #1: (No Response)

Reviewer #3: All comments have been addressed

2. Is the manuscript technically sound, and do the data support the conclusions?

Reviewer #1: (No Response)

Reviewer #3: Yes

3. Has the statistical analysis been performed appropriately and rigorously? 

Reviewer #1: (No Response)

Reviewer #3: Yes

4. Have the authors made all data underlying the findings in their manuscript fully available?

Reviewer #1: (No Response)

Reviewer #3: No

5. Is the manuscript presented in an intelligible fashion and written in standard English?

Reviewer #1: (No Response)

Reviewer #3: Yes

6. Review Comments to the Author

Reviewer #1: (No Response)

Reviewer #3: (No Response)

7. PLOS authors have the option to publish the peer review history of their article (what does this mean?). If published, this will include your full peer review and any attached files.

Reviewer #1: **Yes: **Kristina Aurelius

Reviewer #3: No

---

## [Editor Report · Acceptance letter]

23 Jun 2024

PONE-D-23-40927R1 

PLOS ONE

Dear Dr. Löfgren, 

I'm pleased to inform you that your manuscript has been deemed suitable for publication in PLOS ONE. Congratulations! Your manuscript is now being handed over to our production team.

Kind regards, 

on behalf of

Dr. Muhammad Shahzad Aslam 

Academic Editor

PLOS ONE